



# The Impact of (bio-)organic substances on the ice nucleation activity of the K-feldspar microcline in aqueous solutions

Kristian Klumpp, Claudia Marcolli, Thomas Peter

Institute for Atmospheric and Climate Sciences, ETH Zurich, Zurich, 8092, Switzerland

*Correspondence to*: Kristian Klumpp (kristian.klumpp@env.ethz.ch)

**Abstract**

Potassium-feldspars (K-feldspars), such as microcline, are considered key dust minerals inciting ice nucleation in mixed phase clouds. Besides the high ice nucleation activity of microcline, recent studies also revealed a high sensitivity of microcline

towards interaction with solutes on its surface. Here, we investigate the effect of organic and bio-organic substances on the ice nucleation activity of microcline, with the aim to better understand the underlying surface interactions. We performed immersion freezing experiments with microcline in solutions of three carboxylic acids, five amino acids and two polyols to represent these compound classes. By means of a differential scanning calorimeter we investigated the freezing of emulsified droplets of microcline suspended in various solutions. Depending on the type of solute, different effects were observed. In the case of

carboxylic acids (acetic, oxalic and citric acid), the measured heterogeneous onset temperatures, $T_{het}$, showed no significant deviation from the behavior predicted by the water activity criterion, $T_{het}(a_w) = T_{melt}(a_w + \Delta a_w)$, which relates $T_{het}$ with the melting point temperature $T_{melt}$ via a constant water activity offset $\Delta a_w$. While this behavior could be interpreted as a lack of interaction of the solute molecules with the surface, the carboxylic acids caused the fraction of heterogeneously frozen water, $F_{het}(a_w)$, to decrease by up to 40% with increasing solute concentrations. In combination, unaltered $T_{het}(a_w)$ and reduced $F_{het}(a_w)$

suggest that active sites were largely deactivated by the acid molecules, but amongst those remaining active are also the best sites with the highest $T_{het}$. A deviation from this behavior is citric acid, which showed not only a decrease in $F_{het}$, but also a decrease in $T_{het}$ of up to 4 K for water activities below 0.99, pointing to a depletion of the best active sites by interactions with the citrate ions. When neutralized solutions of the acids were used instead, the decrease in $F_{het}$ became even more pronounced. The slope of $T_{het}(a_w)$ was different for each of the neutralized acid solutions. In the case of amino acid solutions, we found a

decrease in $T_{het}$ (up to 10 K), significantly below the $\Delta a_w$-criterion, as well as a reduction in $F_{het}$ (up to 60%). Finally, in case of the investigated polyols, no significant deviation of $T_{het}$ from the $\Delta a_w$-criterion was observed, and no significant deviation of $F_{het}$ in comparison to a pure water suspension was found. Furthermore, we measured the effects of aging on the ice nucleation activity in experiments with microcline suspended in solutions for up to seven days, and tested the reversibility of the interaction with the solutes after aging for 10 days. For citric acid, an ongoing irreversible degradation of the ice nucleation activity

was observed, whereas the amino acids showed completely reversible effects. In summary, our experiments demonstrate a remarkable sensitivity of microcline ice nucleation activity to surface interactions with various solutes, underscoring the importance of the history of such particles from source to frozen cloud droplet in the atmosphere.





## 1 Introduction

The formation of ice has various implications on key characteristics of clouds. The presence of ice significantly influences
cloud lifetime, the formation of precipitation, cloud chemistry and radiative properties (Lohmann, 2006; Field and Heymsfield,
2015; Mülmenstädt et al., 2015; IPCC, 2013). Numerous aspects of ice formation within clouds are still unclear, necessitating
further research to elucidate the different ice nucleation mechanisms.

Ice formation takes place via homogenous or heterogeneous ice nucleation in supercooled cloud droplets. Homogeneous nu-
cleation occurs spontaneously as a stochastic process when micrometer-sized pure liquid water droplets supercool to 235–238
K (Sassen and Dodd, 1987; Heymsfield and Sabin, 1989) and can be described by classical nucleation theory (e.g. Lohmann
et al., 2016; Ickes et al., 2015). In aqueous solutions, the thermodynamic formulation by Koop et al. (2000) describes the
homogeneous freezing point curve as a function of water activity, $a_w$ of the solution, i.e. the ratio between the water vapor
pressure of the solution and of pure water under the same conditions:

$$T_{\text{hom}}(a_w) = T_{\text{melt}}(a_w + \Delta a_{w,\text{hom}}) \tag{1}$$

This relationship is termed the "water-activity criterion" or "$\Delta a_w$-criterion for homogeneous ice nucleation", which states that,
owing to the thermodynamic properties of water and irrespective of the nature of the solute, the homogeneous freezing point
curve can be constructed from the melting point curve, $T_{\text{melt}}$, by a constant water activity shift $\Delta a_{w,\text{hom}}$ (Koop and Zobrist,
2009). This approach has a broad empirical basis and theoretical underpinning (Baker and Baker, 2004). In contrast, hetero-
geneous ice nucleation is induced by solid particles present in the water or solution, so called ice nucleating particles (INPs).
More specifically the nucleation takes place on active sites, which are preferred locations for ice nucleation (IN) with areas of
10-50 nm$^2$ based on estimates from classical nucleation theory (Vali, 2014; Vali et al., 2015; Kaufmann et al., 2017; Holden
et al., 2019). Depending on the circumstances of ice nucleation, different modes of IN are possible: (i) Immersion freezing
describes the freezing process when an INP is immersed in a supercooled droplet. (ii) Condensation freezing is assumed to
occur when IN coincides with cloud droplet activation (Vali et al., 2015; Kanji et al., 2017). (iii) Contact freezing may occur
when nucleation is initiated by an INP that penetrates the droplet surface, e.g. through a collision (Durant and Shaw, 2005;
Shaw et al., 2005; Nagare et al., 2016). (iv) Finally, deposition nucleation is the term for a freezing process happening without
the liquid water phase involved and therefore assuming a direct transition of gaseous water to ice on an INP surface. The latter
is currently questioned by Marcolli (2014), who suggested water vapor condensation in pores with subsequent pore water
freezing (termed pore condensation and freezing, PCF) to be the dominant nucleation process at water subsaturated atmos-
pheric conditions.

Since the introduction of the $\Delta a_w$-criterion by Koop et al. (2000) for homogeneous freezing, numerous studies found that $\Delta a_w$
= const. might under certain circumstances also be applicable to heterogeneous nucleation in immersion freezing mode occur-
ring in the presence of solutes (Zobrist et al., 2008a; Knopf and Alpert, 2013; Koop and Zobrist, 2009; Rigg et al., 2013), albeit



with $\Delta a_{w,\text{het}} < \Delta a_{w,\text{hom}}$. The relation $\Delta a_{w,\text{het}}$ = const. suggests that the freezing point depression is merely determined by the influence of the solute on $a_w$, whereas specific interactions between the solutes and the INP surface are absent.

Among the particles that are considered important as atmospheric INPs, mineral dust particles are the best-established particle type. The ability of mineral dust particles to nucleate ice depends on their mineralogical composition. Among the minerals present in transported dusts (Reid et al., 2003; Murray et al., 2012; Boose et al., 2016), quartz, feldspars, and clay minerals such as kaolinite, montmorillonite and illite have been found to be IN active (Hoose and Möhler, 2012; Pinti et al., 2012; Hiranuma et al., 2015a; Kumar et al., 2019a; 2019b; Harrison et al., 2019), while other frequent minerals like calcite, dolomite, and micas exhibit negligible IN activity (Kaufmann et al., 2016; Kumar et al., 2019b). Atkinson et al. (2013) emphasized the exceptionally high IN efficiency of K-feldspars compared to other minerals. Indeed, several studies confirmed that K-feldspars nucleate ice at higher temperatures with higher activated fractions than (Na-Ca)-feldspars (Peckhaus et al., 2016: Kumar et al., 2018; Kaufmann et al., 2016; Harrison et al., 2016). But also within the group of K-feldspars, IN activities differ, with microcline being IN active at higher temperatures than orthoclase, adularia, and sanidine (Kaufmann et al., 2016; Welti et al., 2019). Despite the high proportion of feldspars in the Earth's crust, they make up only 13 % of the airborne dust aerosol by mass (Murray et al., 2012). However, considering that particles above 1 µm in diameter are typically aggregates of different minerals, even minor contributions of microcline can determine the IN characteristics of a particle.

Yet, ice nucleation by mineral dusts, which mostly nucleate ice below 255 K (Murray et al., 2012; Atkinson et al., 2013; Kaufmann et al., 2016; Boose et al., 2016), fails to explain the occurrence of ice in clouds at even higher temperatures. Therefore, in recent years, biological particles gained attention as INPs, because of their ability to nucleate ice well above 255 K (Hoose et al., 2010; DeMott and Prenni, 2010; Conen et al., 2011; Pummer et al., 2012; O'Sullivan et al., 2014, 2015; Twohy et al., 2016; Stopelli et al., 2017; Cascajo-Castresana et al.,2020; Huang et al., 2021). Biological particles are of very diverse nature and include bacteria, pollen, spores, and fragments of plants and animals (Murray et al., 2012; Morris et al., 2013; Després et al. 2012; Kanji et al., 2017). The IN active entities can be traced back to macromolecules that include polysaccharides (Steinke et al., 2020) and cellulose (Hiranuma et al., 2015b; 2019), lignin (Bogler and Borduas-Dedekind, 2020; Steinke et al., 2020), humic or fulvic substances (Fornea et al., 2009; Wang and Knopf, 2011; Borduas-Dedekind et al., 2019) and proteinaceous material (Huang et al., 2021). Among these macromolecules, IN activity strongly varies. The ice-active proteins expressed by *Pseudomonas Syringae* may nucleate ice up to 271 K, but only when they are arranged on the outer cell membrane of the intact cell (Maki et al., 1974; Schnell and Vali, 1976; Govindarajan and Lindow, 1988). Freezing at similarly high temperatures has also been observed for related bacterial species and some fungi, most probably also originating from proteinaceous nucleation sites (Lindow et al., 1978; Kim et al., 1987). Yet, even proteins that are not expressed for the purpose to nucleate ice, such as the iron-storing protein apoferritin, proved ice-active up to 269 K when clustered together in aggregates (Cascajo-Castresana et al., 2020). Peat soil, fulvic acids, and aquatic dissolved organic matter showed average freezing temperatures between 260–265 K (Fornea et al., 2009; Borduas-Dedekind et al., 2019). Ice nucleation by cellulose only becomes





relevant at 252 K (Hiranuma et al., 2015b; 2019), although freezing onsets may reach as high as 269 K. For lignin, freezing onsets were observed below 263 K with 50 % frozen fraction reached only at 254 K (Bogler and Borduas-Dedekind, 2020).

Biogenic material is inferred to increase the ice nucleation ability of soil dust compared with desert dust since various studies showed that the IN activity of collected soil dust samples decreases when the samples are treated with heat or hydrogen per-
oxide to remove (bio-)organic residues from the dust samples (Conen et al., 2011; Tobo et al., 2014; O'Sullivan et al., 2018; Paramonov et al., 2018). Similarly, glacial outwash sediments exhibited high IN activity, most probably stemming from the presence of small amounts of biogenic material, since treatment with hydrogen peroxide reduced the ice nucleation active site densities to values found for illite NX (Tobo et al., 2019), often taken as a proxy for mineral dust (Hiranuma et al., 2015a). However, the treatment of bulk samples with heat or hydrogen peroxide does not provide insights whether synergistic effects
between (bio-)organic substances and mineral dust particles are involved in the high IN activity of soil dusts. Such effects have been suggested by O'Sullivan et al. (2014) as their soil dust samples froze at higher temperatures than either Humic-Like Substances (HULIS) or mineral dust references by themselves. Only a few studies have investigated the mixing state of mineral dust and biogenic material and its relevance for the IN activity. Pratt et al. (2009) found external mixtures of mineral dust and biological particles using aircraft aerosol time-of-flight spectrometry applied to cloud ice-crystal residues, yet, they were IN
active only at quite low temperatures (239–242 K). Conversely, single particle analysis of agricultural harvesting emissions revealed besides pure dust and biological particles also a considerable fraction of dust particles mixed with organic or biolog-ical material (Suski et al., 2018). Finally, O'Sullivan et al. (2016) found that ice-nucleating proteins from *Fusarium avenaceum*, a common soil-borne fungus, preferentially bind to kaolinite and impart their ice-nucleating properties to it.

Instead of already adhering to mineral dust particles before entering the atmosphere, organic and biological material may attach
to the mineral particle surface also during cotransport in the atmosphere by coagulation with organic particles or condensation of semivolatile organic molecules. The influence of such coatings may depend on various factors such as the thickness and chemical composition of the coating, affecting the nature of the interaction. Recent investigations showed that mineral dust surfaces are sensitive to exposure to inorganic solutes leading to deviation from $\Delta a_{w,\mathrm{het}}$ = const. Whether the solutes enhance or deteriorate the IN activity compared with $\Delta a_{w,\mathrm{het}}$ = const. depends on the mineralogy of the particles, the nature and concen-
tration of the solutes, and the freezing temperature range. Figure 1 provides a schematic overview of the different interactions between solutes and the mineral dust surface in aqueous solutions, which include ion exchange, adsorption, and dissolution of the mineral surface. Panel (a) depicts the case with no specific interactions between the solute and the mineral surface such that $\Delta a_{w,\mathrm{het}}$ = const. is fulfilled. Cation exchange, as depicted in panel (b) may occur for minerals with exchangeable cations and has been shown to decrease the IN activity of feldspars in general (Kumar et al. (2018, 2019b; Whale et al., 2018; Perkins
et al., 2020; Yun et al., 2020). Ammonia and ammonium salts proved to generally increase the IN activity of aluminosilicates, which includes feldspars and clay minerals, most probably through both, ion exchange (panel b) and adsorption (panel c). In case of quartz, enhancing and deteriorating interactions of the solutes with the surface seem to compete as the presence of ammonia decreased the freezing onset temperatures due to dissolution (panel d) of the mineral surface in basic solution, while



the presence of ammonium sulfate has a slight enhancing effect probably due to adsorption (Kumar et al., 2019a; Whale et al.,
130 2018).

To extend these findings, we aim to gain insight into possible interactions between mineral surfaces and (bio-)organic solutes.
We chose microcline to provide the mineral surface as this feldspar showed on one hand, a high ice nucleation activity, while
on the other hand, it proved to be highly sensitive to the presence of solutes. The organic solutes were selected to represent
typical functional groups present in organic and proteinaceous materials. The hydroxyl groups of polyols feature the prevailing
functional group of polysaccharides and cellulose and are also frequent in lignin, fulvic, and humic acids. We chose glycerol
and heptane-1,7-diol to represent these types of organics. To represent the carboxyl groups of fulvic, and humic acids, we
chose a monovalent (acetic acid), a bivalent (oxalic acid) and a trivalent organic acid (citric acid). The number of carboxyl
groups in an organic acid influences the complexation of $Al^{3+}$ present in feldspars, in turn modulating the ability of the organic
acid to interact with the mineral surface. Moreover, as the strongest carboxylic acid, oxalic acid is supposed to have the most
pronounced effect on feldspar dissolution because of its acidity, which was shown to deteriorate the IN activity of microcline
in the presence of $H_2SO_4$ (Kumar et al., 2018). Finally, with amino acids, we test how interactions between the microcline
surface and the building blocks of proteins influence the IN activity. In solution, amino acids are present as zwitterions with
$-COO^-$ carrying the negative and $-NH_3^+$ the positive charge (Fig. A1 in the appendix). Both groups have the potential to
interact with the microcline surface. The similarity of $-NH_3^+$ with $NH_4^+$ and $NH_3$ raises the question whether such interactions
may enhance the IN activity of feldspars. We chose glycine, l-alanine, l-serine, l-lysine, and l-glutamine to cover relevant
functionalities found in proteins (see Table 1 for structures and properties). Glycine as the simplest amino acid provides the
basic case of no additional functionalities in the side chain. L-alanine possesses an aliphatic tail as the side chain providing the
case without the possibility for polar interactions. With a hydroxyl side chain, serine is capable of additional polar interactions.
L-lysine and l-glutamine possess longer side chains with additional amino and amide functional groups, respectively, that may
interact with the microcline surface. With the amino group, l-lysine has an additional basic group increasing the pH of the
solution to the basic range. Furthermore, all selected amino acids are sufficiently water soluble to potentially modify the mi-
crocline surface distinctly. Besides representing functionalities found in biogenic material, the investigated organic acids are
oxidation products of a lot of organic substances, polyols are intermediate products before further oxidation yields carboxylic
acids, and amino acids were detected in the atmosphere in various locations (Barbaro et al. 2011; 2015).

Using a DSC setup, we will present results from immersion freezing experiments of microcline suspensions dispersed as
emulsified solution droplets.





## 2 Methods

### 2.1 Particle size distribution and surface area of the microcline powder sample

For the herein described experiments, two stones of microcline were used, both from North Macedonia provided by the Institute of Geochemistry and Petrology of ETH Zurich. Both stones were milled with a tungsten carbide disc mill resulting in a fine powder. The first stone is identical to the one used by Kumar et al. (2018). The second stone was milled in October 2019 following the same protocol as for the first stone. X-ray diffraction followed by Rietveld analysis of powder samples yielded the following composition: (86.33 ± 1.7) % microcline, (6.18 ± 0.72) % orthoclase and (7.49 ± 0.48) % albite for the first

sample and (91.9 ± 0.8) % microcline and (8.9 ± 0.4) % albite for the second sample. The number size distributions of the mineral dust particles suspended in water were measured using laser diffraction analysis (see supplementary material).

The specific surface area was determined using dynamic vapor sorption (DVS) analysis yielding a value of 1.6 m²/g for the first and 3.1 m²/g for the second sample.

### 2.2 Immersion freezing experiments of emulsified microcline suspensions in water and aqueous solutions

Immersion freezing experiments were conducted with a Differential Scanning Calorimeter DSC Q10 from TA instruments (see Zobrist et al., 2008a, for more details). Suspensions of microcline (2 wt%) in aqueous (molecular bioreagent water, Sigma Aldrich) solutions containing different organic acids, amino acids and polyols as detailed in Table 1 were prepared. In order to prevent particle aggregation, all samples were sonicated 5 to 10 minutes before experiments were started. After sonication the suspensions were combined with a mixture of mineral oil (95%) and lanolin (5%) (both Sigma Aldrich) at a ratio of 1:4

and emulsified with a rotor stator homogenizer (Polytron PT 1300D with a PT-DA 1307/2EC dispersing aggregate) for 40 s at 7000 rpm for the measurements including the organic acids and the neutralized organic acids (section 3.1.1 and 3.1.2). For all other measurements the procedure was readjusted to 40 s dispersing at 10000 rpm, using a lanolin content of 7 % to obtain a droplet size distribution that peaks between 2–3 µm with respect to number and between 4–12 µm with respect to volume (for more details see Marcolli et al., 2007; Pinti et al., 2012; and Kaufmann et. al., 2016). Occasionally, single large droplets

may form, which appear as spikes in the thermogram upon freezing. These spikes were neglected in the evaluation. 5 to 10 mg of the emulsion were placed in an aluminum pan, hermetically sealed, placed in the DSC, and measured at cooling and heating rates of 1 K/min. To test the stability of the emulsion, some samples were subjected to three freezing cycles following the procedure introduced by Marcolli et al. (2007) with a first and third cycle performed with a cooling rate of 10 K/min as control cycles. Emulsions were freshly prepared before each experiment. Every experiment was repeated at least once with a freshly

prepared suspension. For evaluation of the freezing onset temperatures ($T_{het}$ and $T_{hom}$), the heterogeneously frozen fraction ($F_{het}$) and the melting temperature $T_{melt}$, freezing and cooling cycles run at 1 K/min were used.

Figure 2 shows a typical DSC thermogram of an aqueous microcline emulsion. Depicted is the heat flow of the sample as a function of temperature. Freezing events are accompanied with a release of latent heat resulting in a peak in the thermogram. The first peak at higher temperatures arises from heterogeneous freezing caused by microcline in the emulsion droplets. The

second peak is due to the homogeneous freezing of the remaining droplets, which either contain no microcline particles or



whose microcline particles lack active sites. Our key parameters to analyze the thermograms are the onset temperatures of these peaks, $T_{het}$ and $T_{hom}$. Conversely, for the melting temperature ($T_{melt}$) the maximum of the melting peak is used. The heat released during the experiment is approximately proportional to the volume of water that froze in the sample with a minor deviation from this proportionality arising from the temperature dependence of the freezing enthalpy (Speedy, 1987; Johari et al., 1994). We quantify the heat release by taking the integral over the thermogram and refer to the proportion of the heterogeneous peak integral as the heterogeneously frozen fraction ($F_{het}$). See Fig. 2 for a detailed description of the evaluation of $T_{het}$, $T_{hom}$ and $F_{het}$, and Table 2 for a summary of the uncertainty for each parameter.

## 2.3 Neutralization of carboxylic acid solutions with ammonia

In order to discriminate between the influence of the carboxylate anion and the cation ($H^+$ providing acidity to the solution) of the dissociated carboxylic acids on microcline, suspensions were prepared with neutralized solutions of acetic acid, oxalic acid and citric acid. Acid solutions were neutralized by adding aqueous ammonia (25% (aq), Merck) solution until a pH of 6–8 was reached (tested by pH test strips, Machery-Nagel). For neutralization we deliberately chose ammonia solution and no other alkaline substance, because the influence of ammonia on microcline has been well characterized by Kumar et al. (2018).

## 2.4 Immersion freezing experiments as a function of aging

Microcline suspensions in citric acid (1 wt%), l-alanine (1 wt%) and l-lysine (0.05 wt%) solutions were prepared (see Table 1 for information on supplier and purity) and investigated by DSC over the course of seven days to test the long-term influence of the solutes on the microcline surface. Measurements were performed directly after preparation and on the first, fourth and seventh day after preparation. The suspensions were sonicated 5 min prior to each measurement in order to re-suspend settled particles and to decrease particle aggregation.

## 2.5 Reversibility of interactions between microcline and solutes tested in immersion freezing experiments

Microcline suspensions in citric acid (10 wt%), l-alanine (10 wt%) and l-lysine (2 wt%) solutions were prepared and aged in a sealed glass vial at room temperature for 10 days. After aging, all suspensions were centrifuged (3 min at 600 rpm), the supernatant was removed and the remaining sample washed with water. This procedure was repeated 5 times. The washed particles were re-suspended in water. Finally, immersion freezing experiments were conducted with these suspensions to compare the freezing properties of the aged microcline dust with freshly prepared suspensions in water.

# 3 Results and Discussion

## 3.1 Carboxylic acids

### 3.1.1 Freshly prepared carboxylic acid samples

Figure 3 shows the melting temperatures ($T_{melt}$) and homogeneous and heterogeneous freezing onset temperatures ($T_{hom}$ and $T_{het}$) as a function of water activity ($a_w$) for microcline (2 wt%) in aqueous acetic, oxalic, and citric acid solutions. Analogous





to the procedure described in Kumar et al. (2018), water activities were determined from the melting temperatures obtained from the DSC thermograms using the parameterization introduced in Koop et al. (2000). Therefore, the plotted melting temperatures perfectly align on the ice melting curve (dash-dotted line). Furthermore, we assume the water activities at the freezing

temperatures to be identical to those determined at the melting temperature. This approximation may lead to errors, since water activities show a dependence on temperature, which may explain some of the deviations of the measured homogeneous freezing points from the homogeneous freezing curve (dotted line). Note that these temperature dependencies differ depending on the solute and are often small (Zobrist et al., 2008b; Ganbavale et al., 2014). The homogeneous as well as the heterogeneous freezing curves (black solid and dotted lines, respectively) were obtained by constant shifts in water activity, $\Delta a_{w,\mathrm{hom}}$ and

$\Delta a_{w,\mathrm{het}}$, applied to the ice melting curve (Koop et al., 2000; Zobrist et al., 2008a). Both, $\Delta a_{w,\mathrm{hom}}$ and $\Delta a_{w,\mathrm{het}}$ were calculated to match the average onset temperatures of all measurements performed with the two pure microcline samples (2 wt%) at a water activity of $a_w = 1$, which yielded $\Delta a_{w,\mathrm{hom}} = 0.295$ and a heterogeneous offset $\Delta a_{w,\mathrm{het}} = 0.191$. Note that $\Delta a_{w,\mathrm{hom}}$ is slightly lower than the values 0.305 and 0.313 determined by Koop et al. (2000) and Koop and Zobrist (2009), respectively, in order to take the characteristics of our DSC experiment (droplet size distribution and uncertainty in temperature calibration) into account.

The solid black curve represents the heterogeneous freezing onset temperature expected in the absence of specific interactions between the ice nucleation active surface and the solute, i.e. $T_{\mathrm{het}}(a_w) = T_{\mathrm{melt}}(a_w + \Delta a_{w,\mathrm{het}})$, assuming that $T_{\mathrm{het}}$ is given solely by the freezing point depression due to the decreased water activity as is depicted in Fig. 1a.

For the experiments with the freshly prepared carboxylic acids and the neutralized acid suspensions the first sample was used. For this sample, $T_{\mathrm{het}}$ and $F_{\mathrm{het}}$ of 2 wt% microcline in pure water are 251.7 K and 0.69, respectively, while it was 250.9 K and

0.74 for the second sample, which was used for the aging and recovery experiments and the measurements with all other solutes. The values measured for pure microcline in this study are slightly lower than those found by Kumar et al. (2018; $T_{\mathrm{het}}$ = 251.9 K/ $F_{\mathrm{het}}$ = 0.75).

Figure 3a shows that the heterogeneous freezing temperatures of microcline in aqueous acetic and oxalic acid solutions fulfil $T_{\mathrm{het}}(a_w) = T_{\mathrm{melt}}(a_w + \Delta a_{w,\mathrm{het}})$ at all measured concentrations within experimental uncertainty. Citric acid shows a deviation

towards lower temperatures of up to 4 K for $a_w < 0.99$, i.e. $\Delta a_{w,\mathrm{het}} \neq \mathrm{const}$. The carboxylic acids investigated here show neither the clear enhancement of ammonia containing solutes nor the clear decrease of inorganic "non-ammonia" solutes, which were found by Kumar et al. (2018). However, there is a clear decrease in the heterogeneously frozen fraction for all carboxylic acids as shown in Fig. 3b, pointing to a loss of ice active sites in acid solutions (high $H^+$ concentration), as it was observed by Kumar et al., (2018) for microcline in sulfuric acid solutions and by Yun et al. (2021) in carboxylic and inorganic acid ($HNO_3$ and

HCl) solutions with a trend of decreasing IN activity with decreasing pH (Yun et al. (2021). For acetic acid, as the weakest acid, the decrease in $F_{\mathrm{het}}$ is indeed the least and levels off after a first decline.





### 3.1.2 Neutralized carboxylic acid samples

To test the hypothesis that high concentrations of $H^+$ (low pH) cause the loss of microcline IN activity, we neutralized the carboxylic acid solutions with ammonia to a pH of 6 to 8. We chose ammonia as the neutralizing base, as Kumar et al. (2018)
have shown that ammonia does not deteriorate the ice nucleation activity of microcline. On the contrary, it can even enhance it.

Figure 4 shows the results of these experiments, i.e. the melting ($T_{melt}$) and freezing onset temperatures ($T_{het}$, $T_{hom}$) of microcline in all three aqueous dicarboxylic acid solutions neutralized to pH 6-8 with ammonia. Similarly to the ammoniated solutions investigated by Kumar et al. (2018), an increase in $T_{het}$ is observed for water activities close to 1 and a decrease for lower water
activities. For citric and oxalic acid, the enhancement is restricted to a very narrow concentration range ($1 > a_w > 0.99$) and reverts to a clear decline for higher concentrations that exceeds the one observed in the respective pure acid solutions shown in Fig. 3. None of the neutralized carboxylic acids satisfies the $\Delta a_w$-criterion, i.e. $T_{het}(a_w) \neq T_{melt}(a_w + \Delta a_{w,het})$, and all of them behave differently from one another. Similarly to the pure acids and in clear contrast to ammonia, the neutralized acid solutions show a strong decrease in $F_{het}$ towards lower water activities, which is the strongest for oxalic acid, followed by citric acid,
and the least for acetic acid. This dependence on the specific neutralized acid suggests the presence of interactions between the carboxylic acid or the carboxylate anions and the microcline surface.

The differences between the tested carboxylates can be explained by the complexation of $Al^{3+}$ with the deprotonated carboxylic acids. Indeed, bidental and tridental carboxylic anions have the potential to complex aluminum, while the ability of monodental carboxylates was found to be negligible (Stoessell and Pittman, 1990). Several studies have shown the ability of oxalate to
complex $Al^{3+}$ (Bevan and Savage, 1989; Stoessell and Pittman, 1990; Min et al. 2015). With three acid groups, citric acid might be a similar or even stronger complexing agent. Conversely, adsorption of acetic acid is weaker, as it carries only one carboxylic acid group. This explains why the trend of $T_{het}$ (see Fig. 4a) for microcline in neutralized ammonium acetate solutions (blue squares) is similar to the one for ammonia solutions (cyan diamonds), since ammonia or the ammonium ion dominate surface interactions with respect to acetate. Yet, based on experiments with freshly prepared microcline suspensions, it remains
unclear whether carboxylate complexation leads to enhanced dissolution of microcline or whether it blocks IN active sites of microcline through adsorption of the carboxylic anions on the mineral surface.

### 3.1.3 Aging experiments

In order to detect changes occurring during aging within atmospherically relevant timescales, we performed experiments where microcline (2 wt%) was immersed in the solutions over the course of seven days (Fig. 5) with measurements on days 1, 4 and
7. Figure 5a shows no significant change of $T_{het}$ and $F_{het}$ over seven days when microcline is suspended in pure water. For the same type of experiment, yet with a different microcline sample, Kumar et al. (2018) observed a decrease in $F_{het}$ by one third after one day but no drop in $T_{het}$. Conversely, Peckhaus et al. (2016) found a decrease of 2 K in the median freezing temperature



after five months for their microcline sample. These differences in IN activity after aging highlight differences between microcline samples which may be explained by minor contributions from other minerals that differ from sample to sample and differences in microtexture (Whale et al., 2017).

Figure 5b shows the DSC thermograms of microcline suspended in citric acid solution (1 wt%, $a_w \approx 0.995$). The aging measurements in Fig. 5b show a further decrease in $F_{het}$ and $T_{het}$ evident from the measurements on day 4. There is also a visible change in the shape of the thermogram (the maximum of the heterogeneous freezing peak shifts to lower temperatures). This shows that surface modifications are ongoing leading to progressive loss of IN activity over time. Figure 6 (light green curve) shows the thermogram after 10 days of aging in 10 wt% citric acid solution ($a_w \approx 0.984$) and subsequent washing with pure water. The heterogeneous freezing peak of microcline after this procedure has almost completely disappeared. This illustrates the irreversibility of the surface modification induced through citric acid after 10 days.

The decrease in $F_{het}$ with decreasing $a_w$, the continuous activity loss over time and the irreversibility suggests that there is a permanent destruction of active sites, which was also shown by Kumar et al. (2018) for sulfuric acid acting on microcline. As carboxylic and sulfuric acid both release $H^+$ to the solution, we conclude that high $H^+$ concentrations (low pH) lead to the irreversible loss of ice-active sites of microcline through dissolution as depicted in Fig. 1d.

Figure 5c shows DSC thermograms of microcline suspended in neutralized citric acid solution (1 wt%; $a_w \approx 0.993$). At this concentration, the IN activity of freshly prepared microcline even exceeds the curve of $\Delta a_{w,het}$ = const., which we ascribe to the presence of ammonia. Contrary to the aging in a citric acid solution the IN activity remained stable over the observed time period, supporting the hypothesis that high $H^+$ concentrations are the cause for the permanent loss of ice-active surface sites through dissolution. This is reinforced by Fig. 6 (dark green curve), which shows that even if aged in higher concentration of neutralized citric acid (10 wt%; $a_w \approx 0.975$), the IN activity is restored after washing and resuspension of the sample in pure water. This reversibility indicates that the strong decrease in IN activity during exposure (Fig. 4; $T_{het}$ = 235.9 K and $F_{het}$ = 0.16) to neutralized acid solutions is caused by blocking of IN active sites through surface complexation rather than loss of active sites through dissolution (adsorption mechanism in Fig. 1c). The blocking scenario is in agreement with findings by Bevan and Savage (1989) who showed that suspending K-feldspars in neutralized oxalate solutions increases surface dissolution only weakly compared with strongly increased dissolution rates induced by high $H^+$ concentration. The heterogeneous freezing onset temperature of microcline in neutralized citric acid solution is even higher than in pure water most likely caused by small amounts of ammonia that are still present in the sample, since ammonia has an enhancing effect on IN activity of microcline already at very low concentrations (Kumar et al., 2018).

## 3.3 Polyols

Figure 7 shows melting ($T_{melt}$) and freezing onset temperatures ($T_{het}$, $T_{hom}$) of microcline in aqueous glycerol and 1,7-heptanediol solutions. Over the whole measured concentration range, both solutes show full compliance with $T_{het}(a_w) = T_{melt}(a_w + \Delta a_{w,het})$. Similarly, no significant deviation of $F_{het}$ compared to the pure microcline sample was observed. This suggests that





the interaction of these molecules with the microcline surface must be very similar to the ones of water molecules, namely a

fast dynamic exchange of molecules at the microcline surface corresponding to the case shown in Fig. 1a. As a result, neither

an enrichment nor a depletion of the polyol molecules occurs on the microcline surface. In consequence, these molecules affect

the IN activity only via the reduction of the water activity to the extent predicted by the $\Delta a_w$-criterion. These findings are in

accordance with Yun et al. (2021), who found no significant effect of polyols on the median freezing temperature of potassium-

rich feldspar.

### 3.4 Amino acids

### 3.4.1 freshly prepared amino acid samples

Figure 8 shows melting ($T_{\text{melt}}$) and onset freezing temperatures ($T_{\text{het}}$, $T_{\text{hom}}$) of microcline in aqueous solutions of glycine, L-

alanine, L-serine, L-lysine and L-glutamine. All five tested amino acids show a clear and relatively uniform decrease in $T_{\text{het}}(a_w)$

of up to 9.1 K below $T_{\text{melt}}(a_w + \Delta a_{w,\text{het}})$. Similarly to the carboxylic acids and the neutralized carboxylic acids, $F_{\text{het}}$ strongly

decreases with decreasing water activity for L-lysine, while the other amino acids show only a moderate decrease in $F_{\text{het}}$.

### 3.4.2 Aging experiments

Figure 9 shows the DSC thermograms of microcline suspended in solutions of l-alanine (1 wt%; $a_w \approx 0.994$) and l-lysine

(0.05 wt%; $a_w \approx 0.9985$) over the course of seven days, measured on days 1, 4 and 7. We chose l-lysine, because it showed the

strongest decrease in the fresh measurements and l-alanine as a representative of the other amino acids. The amino acid con-

taining suspensions show more variability in $F_{\text{het}}$ and $T_{\text{het}}$ over the course of seven days compared to the pure microcline

sample, but no clear tendency during the observed timespan. Furthermore, Fig. 10 (magenta and orange curves) shows that

after the procedure of 10 days aging in more concentrated solutions (l-alanine 10 wt%, $a_w \approx 0.977$; l-lysine 2 wt%; $a_w \approx 0.993$)

and subsequent washing, the thermograms show no significant difference from the pure microcline sample after the same

procedure. Despite the strong response observed in the freezing experiments in freshly prepared suspensions, the aging exper-

iments show that ice nucleation is directly affected by the presence of the amino acids with no additional aging effect. This

indicates that the interaction of the amino acids with the microcline surface is reversible once the solutions are removed or

highly diluted. This fits to an adsorption mechanism depicted in Fig. 1c. Due to the zwitterionic properties of the amino acids

(Fig. A1) the positively charged $-\text{NH}_3^+$ functional groups can establish very stable interactions with the negatively charged

microcline surface, which explains the strong response of $T_{\text{het}}$. This may also explain the relatively strong reaction of l-lysine

(also in $F_{\text{het}}$), because with its additional amino group, it is able to form an additional positively charged $-\text{NH}_3^+$ functional

group, by receiving $\text{H}^+$ from the surrounding water. Contrary to the case of ammonia, this strong adsorption goes along with a



decrease in IN activity, since the amino acids – in contrast to ammonia – do not exhibit positively polarized H-atoms ready for

hydrogen bonding to the water oxygen.

## 4 General discussion and implications

Depending on the solute, different influences on the freezing onset temperatures and the heterogeneously frozen fraction were observable. Overall, these investigations confirm the high sensitivity of the IN activity of microcline towards solutes as was already shown for inorganic salts (Kumar et al., 2018; Whale et al. 2018; Yun et al., 2020) and sulfuric acid (Kumar et al.,

350  2018).

The here investigated carboxylic acids and the sulfuric acid investigated by Kumar et al. (2018) show similar effects on the IN activity of microcline, namely: $T_{het}$ following $\Delta a_w$-criterion; $F_{het}$ decreasing with increasing concentration together with irreversibility of the decrease. This supports the conclusion that the effect of acids originates primarily from the surface interaction with H$^+$ leading to dissolution and only secondarily from interactions with the anions. The influence of anions becomes visible

in experiments under neutral pH, evidencing the complexation of Al$^{3+}$ in the surface layers of the mineral. Bidentate complexes of oxalate (and dicarboxylic acids in general) with Al$^{3+}$ have been reported to accelerate the dissolution of the feldspar anorthite, which is the Ca$^{2+}$ endmember of the plagioclase series (Min et al., 2015). Yet, because the higher Si : Al ratio of microcline (3:1) compared with anorthite (1:1) leads to a higher number of Si–O–Al compared with Al–O–Al bonds (Oelkers and Schott, 1995; Oelkers et al., 2009), dissolution is much lower in microcline because the aluminum ions

are more tightly bound within the crystal lattice through Si-O-Al bonds compared with Al-O-Al bonds. A slight effect of carboxylic anions on the dissolution of feldspars has been shown by Bevan and Savage (1989) who independently varied pH and oxalic acid concentrations in K-feldspar suspensions and found strongly enhanced surface dissolution when pH was decreased compared with only a small increase of dissolution when the oxalate concentration was increased. We therefore conclude that, on the timescales of our experiments, bidentate and tridentate carboxylates decrease the IN activity by blocking

active sites through surface adsorption rather than by dissolving them. Surface complexation is smaller for acetic acid since as a monocarboxylic acid, it cannot form bidentate complexes with Al$^{3+}$. We also speculate that complexation with carboxylic acids would decrease the IN activity of microcline irreversibly on longer timescales. We assume that in the case of feldspars with lower Si to Al ratio, surface complexation would immediately promote dissolution. Under strongly acidic conditions (pH 1), Bevan and Savage (1989) found decreased dissolution rates of K-feldspar in the presence of oxalic acid. This indicates that

H$^+$ catalyzed dissolution can even be inhibited instead of promoted by oxalate complexation. Yet, at such high H$^+$ concentrations, dissolution dominates complexation by far leading to a net loss of active sites within the timescale of our experiments.

The investigated amino acids show a decrease in $T_{het}$ and $F_{het}$, which is strongest in the case of L-lysine. Due to the zwitterionic nature of amino acids a strong polar interaction with the hydroxyl groups on the mineral surface is possible. Periodic density functional theory calculations have shown that glycine, the simplest amino acid, can interact with up to three surface silanol

groups of microcline (Hu et al., 2020). The resulting adsorption of the molecules alters the surface properties and therefore





influences the IN activity. In the case of L-lysine the additional, positively charged $-NH_3^+$ group is expected to additionally enhance the adsorption on the negatively charged surface of microcline ($pH_{PZC} < 2$; Vidyadhar and Hanumantha Rao, 2007), resulting in even lower $F_{het}$. Hedges and Hare (1987) showed in experiments that basic (positively charged) amino acids adsorbed much better on clay mineral surfaces than neutral and acidic amino acids, confirming a dependence of surface ad-

sorption on the surface charge of the mineral.

The effect of polyols on IN activity aligns with the $\Delta a_w$-criterion, which is in accordance with Yun et al. (2021) and findings from numerous studies which investigated the impact of organic coatings on various mineral dusts (Tobo et al., 2012; Wex et al., 2014 (kaolinite + levoglucosan); Zobrist et al., 2008a; Koop and Zobrist, 2009 (ATD + polyethylene glycol);). Kanji et al. (2019) investigated collected dust particles from the Sahara and Asia coated with secondary organic aerosol from dark ozo-

nolysis of α-pinene. The particles were coated and subsequently the ice nucleation properties were observed in continuous flow diffusion chambers. Even though the SOA coating contained also carboxylic acid functional groups, no decrease in the IN activity of the dust was found in this study. However, as the share of microcline in the samples was not stated, the presence of other IN active mineral components that do not respond as sensitive to organic acids as microcline may offer an alternative explanation for this finding. Apart from that, the $H^+$ concentration and the competition of $H^+$ with other complexing agents

present in SOA also play a role for how accessible the mineral surface is for $H^+$ ions of coated particles.

None of the here presented (bio-)organic substances enhanced the IN activity of microcline, which rules them out as a cause for enhanced IN activity of soil dusts. Furthermore, interactions between the mineral surface and the most prevalent organic functional groups of the organic aerosol fraction leave the IN activity unchanged or even decrease it. These findings cast doubt on the conception that molecular interactions of organic molecules with the mineral surface may explain the enhanced IN

activity of soil dusts, and leaves ammonia and ammonium as the only identified agents that are able to increase the IN activity of mineral particles. Actually, ammonia is an omnipresent fertilizer of soils that indeed might have a share in increasing the IN activity of soil dusts. In addition, ammonia will be removed by heating or $H_2O_2$ treatment, which are often used to infer the presence of (bio-)organic components (Du et al., 2017; O'Sullivan et al., 2014; Hill et al., 2016), which makes it an interfering agent in those tests.

Alternatively, or additionally, mineral particles may act as a substrate to order macromolecules such as humic acids, proteins or fragments of even larger systems, thus enhancing their IN activity. In this case, the macromolecules would provide the active sites, which may be ordered to form larger IN active structures on the mineral surface that are able to nucleate ice at higher temperatures. Such an effect due to aggregation on a substrate has been observed in the case of *Pseudomonas syringae,* where the intact outer cell membrane provides a substrate for ordering the IN active proteins into larger aggregates, thus raising

the freezing temperatures by about 5 K (Polen et al., 2016; Zachariassen and Kristiansen, 2000)). Findings of O'Sullivan et al. (2016) have shown that the IN activity of ice-nucleating proteins released from *Fusarium avenaceum* is conferrable to kaolinite dust particles when the protein encounters the mineral, yet, without enhancing its IN activity. Thus, it remains an open question whether synergistic effects between (bio-)organic substances and mineral surfaces are able to enhance the IN activity of either component. Alternatively, biogenic macromolecules like proteins exhibit high freezing temperatures by themselves and their



presence alone may explain the increased IN activity of soil dusts compared with bare mineral dusts. This hypothesis gains relevance considering that also proteins that are not expressed for the purpose to nucleate ice can act as INPs (Cascajo-Castresana et al., 2020), at high temperatures (up to 269 K).

## 5 Conclusions and summary

Immersion freezing experiments of suspended microcline particles in different carboxylic acid, amino acid and polyol solutions showed diverse effects on the IN activity of microcline attributable to their different surface interaction. The two investigated polyols showed agreement (within uncertainty) with the $\Delta a_w$-criterion and no significant change in the heterogeneously frozen fraction $F_{\mathrm{het}}$. The three examined carboxylic acids were also in agreement with the $\Delta a_w$-criterion, except for citric acid concentrations that correspond to $a_w < 0.99$. Yet, all three carboxylic acids cause a significant decrease in $F_{\mathrm{het}}$. When the acids

were neutralized with ammonia, the decrease in $F_{\mathrm{het}}$ was even more pronounced and the three acids yielded trends in $T_{\mathrm{het}}$ different from one another and from the $\Delta a_w$-criterion. The five amino acids showed decreased onset temperatures by up to 10 K below the $\Delta a_w$-criterion as well as decreased $F_{\mathrm{het}}$ values. Furthermore, aging experiments showed that citric acid led to irreversible loss of ice-active sites while aging in neutralized citric acid, l-lysine and l-alanine solutions was completely reversible. This illustrates the different interactions of these solutes with the microcline surface. While acid solutions lead to

irreversible loss of IN activity through dissolution of the microcline surface, the effect of all other investigated species is reversible and can be explained by adsorption leading only to temporary loss of IN activity of microcline. Our investigations show that the IN activity of microcline is highly sensitive to bio-organic solutes in agreement with other studies that showed a generally high sensitivity of microcline with respect to solutes. These findings highlight the need to investigate not only the IN activity of pure minerals but to extend these studies to investigations in the presence of species the minerals are in contact

with throughout their emission and transport in the atmosphere. Furthermore, we cannot explain elevated IN activity of soil dusts based on surface interaction of microcline with the here presented bio-organics.

*Data availability.* The data presented in this publication will be submitted to the ETHZ data repository soon.

*Author contributions.* KK conducted the experiments. KK, CM, and TP contributed to the planning and interpretation of the
experiments. KK prepared the manuscript with contributions from CM and TP.

*Competing interests.* The authors declare that they have no conflict of interest.

*Acknowledgements.* We acknowledge the Swiss National Foundation for financial support (project numbers: 200021_175716). We thank Michael Plötze, Anette Röthlisberger, and Marion Rothaupt for XRD and BET measurements; Philippe Grönquist and Christoper Dreimol providing us with DVS measurements; Marco Griepentrog for milling; Peter Brack for providing the
microcline samples; Ulrich Krieger, Uwe Weers, Nikou Hamzepour, and Marco Vecellio for support in the laboratory (all





ETH). We also thank Silvan von Arx from the Institute of Mechanical Engineering and Energy Technology (Lucerne School of Engineering and Architecture, Lucerne) for providing size distribution measurements (Laser diffraction particle sizer).

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





**Table 1.** Summary of all solutes used. Stated are the supplier of the substance, purity and the concentration range that was used in the experiments.

| Substance | Supplier | Purity | Concentration range | Structure | pKa |
|---|---|---|---|---|---|
| **Carboxylic acids** | | | | | |
| Acetic acid | Fluka | $\geq 99.5\%$ | 0.01 – 15 wt% pH: 3.8 – 2.2[1] | | pKa = 4.756 |
| Citric acid | Sigma Aldrich, Fluka (monohydrate) | $\geq 99.5\%$ | 0.01 – 20 wt% pH: 3.4 – 1.6[2] | | pKa$_1$ = 3.13 pKa$_2$ = 4.76 pKa$_3$ = 6.39 |
| Oxalic acid | Sigma Aldrich | $\geq 99.5\%$ | 0.01 – 4 wt% pH: 3.0 – 0.9[2] | | pKa$_1$ = 1.25 pKa$_2$ = 4.14 |
| **Amino acids** | | | | | |
| L-Alanine | Merck | $\geq 98\%$ | 0.01 – 10 wt% | | pKa$_{COOH}$ = 1.25 pKa$_{NH_3^+}$ = 9.87 |
| L-Glutamine | Merck | $\geq 99\%$ | 0.01 – 2 wt% | | pKa$_{COOH}$ = 2.17 pKa$_{NH_3^+}$ = 9.13 |
| Glycine | Sigma Aldrich | $\geq 99\%$ | 0.1 – 5 wt% | | pKa$_{COOH}$ = 2.21 pKa$_{NH_3^+}$ = 9.15 |
| L-lysine | Sigma Aldrich | $\geq 98\%$ | 0.001 – 10 wt% pH: 10.4 – 11.5 | | pKa$_{COOH}$ = 2.20 pKa$_{NH_3^+}$ = 8.90 pKa$_{NH_2}$ = 10.28 |
| L-serine | Sigma Aldrich | $\geq 99\%$ | 0.02 – 20 wt% | | pKa$_{COOH}$ = 2.21 pKa$_{NH_3^+}$ = 9.15 |
| **Polyols** | | | | | |
| Glycerol | Honeywell | $\geq 99\%$ | 0.5 – 15 wt% | | |
| 1,7-Heptanediol | Sigma Aldrich | $\geq 95\%$ | 0.1 – 10 wt% | | |


---

[1] Calculated using the weak acid approximation $pH = 0.5 * (pKa - \log_{10}(c_{acid}))$ ($c_{acid}$ in mol/l)

[2] Calculated using the semi strong approximation and consideration of the first 2 acidic protons ($c_{acid}$ in mol/l)

$$pH = -\log_{10}\left(\frac{K_a}{2} + \sqrt{\frac{K_a^2}{4} + K_a * c_{acid}}\right)$$





**Table 2.** Summary of statistical uncertainties of measurement parameters. Stated are the average standard deviation of $T_{het}$, $T_{hom}$, $T_{melt}$, and $F_{het}$ determined as the average of the standard deviations from all measurement points shown in Figs. 3, 4, 7, 8 and the maximum standard deviation observed for each measurement parameter.

| Parameter | Average deviation | Maximum deviation |
|---|---|---|
| $T_{het}$ | 0.25 K | 1.29 K |
| $T_{hom}$ | 0.07 K | 0.57 K |
| $T_{melt}$ | 0.06 K | 0.41 K |
| $F_{het}$ | 0.05 | 0.14 |




**Figure 1.** Schematic visualization of a mineral dust particle fully immersed in an aqueous solution droplet. Panels (a) to (d) depict different interactions of solutes on the mineral dust surface. (a) no specific interaction of the solute with the surface, which results in neither enrichment nor depletion of solute on the surface, and thus $T_{het}(a_w) = T_{melt}(a_w + \Delta a_{w,het})$ with $\Delta a_{w,het}$ = const; (b) in the case of ionic solutes, cations located in the surface layer of the mineral can be exchanged with cations of the surrounding solution; (c) the solute adsorbs to the surface leading to a surface enrichment and possibly coverage and blocking of IN active sites; (d) dissolution of the surface layer through reactive solutes (e.g. acids). Interactions depicted in panels (b) to (d) are characterized by $\Delta a_{w,het} \neq$ const, and typically $T_{het}(a_w)$ is lower than expected from the $\Delta a_w$-criterion, i.e. a worsening of ice nucleation ability with increasing solute concentration, i.e. decreasing $a_w$.



**Figure 2:** A typical DSC thermogram, showing freezing onset temperatures $T_{hom}$ and $T_{het}$ constructed with the asymptotes of the homogeneous and the heterogeneous freezing peaks, respectively, and the frozen fractions $F_{hom}$ and $F_{het}$ resulting from the integral between the DSC curve and the blue horizontal line, as the base line. The area underneath the curve corresponds to the frozen water volume (if the temperature dependence of the enthalpy of freezing is neglected; Speedy, 1987; Johari et al., 1994). All depicted DSC thermograms in this study are normalized with respect to their total integral i.e. $F_{het} + F_{hom} = 1$. The homogeneous onset temperature is taken as the separator between the two peak areas. Note that integrals are determined in the time domain (heat flow vs. time) and not in the temperature domain, which is shown here.






**Figure 3:** Freezing experiments with microcline (2 wt%) in aqueous carboxylic acid solutions as a function of water activity. For comparison, microcline 2 wt% in sulfuric acid from Kumar et al. (2018) data are shown as magenta pentagons. (a) Melting temperatures, $T_{melt}$ (open symbols between 265 K and 275 K), heterogeneous freezing onset temperatures (filled symbols) and homogeneous freezing onset temperatures (open symbols between 220 K and 240 K) as a function of water activity, $a_w$. Dash-dotted black line: ice melting point curve; solid black line: heterogeneous freezing curve, $T_{het}(a_w)$; dotted black line: homogeneous freezing curve, $T_{hom}(a_w)$. All points are the average of at least two separate emulsion freezing experiments. Measurements without a repetition are marked with a black star label (see supplemental information for individual DSC curves). (b): Heterogeneously frozen fraction, $F_{het}$, as a function of water activity, $a_w$. Measurement uncertainties are shown for one exemplary data point and for the microcline pure measurement as the spread from minimum to maximum measured values.





**Figure 4:** Compiled freezing experiment results of microcline (2 wt%) in neutralized aqueous carboxylic acid solutions of varying concentration. Cyan diamonds for $NH_3$-solutions from Kumar et al. (2018). Data points and curves are as explained in Fig. 3.



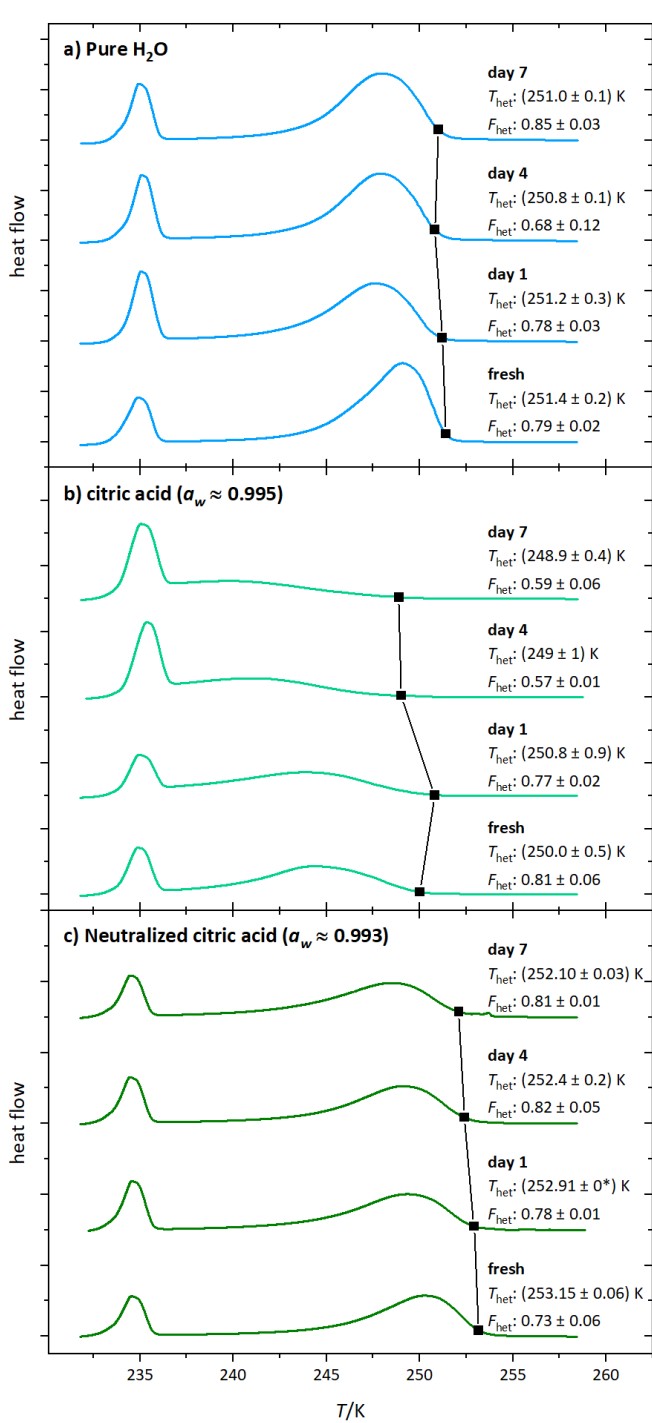

**Figure 5:** DSC thermograms of microcline (2 wt%) in (a) water (b) 1 wt% citric acid ($a_w \approx 0.995$) and (c) 1 wt% citric acid neutralized with $NH_3$ ($a_w \approx 0.993$) measured directly after the preparation (fresh) and on the displayed days (day 1, 4, 7). Black squares mark the heterogeneous freezing onset temperatures. The thermograms are normalized with respect to their total integral. The displayed $T_{het}$ and $F_{het}$ values and the black squares are the average of two measurements. The corresponding uncertainties are the standard deviation (black star label: both measurements yielded the same $T_{het}$).





**Figure 6:** DSC thermograms of microcline (2 wt%) after 10 days of aging in the respective solutions (citric acid: 10 wt%, $a_w \approx 0.984$; neutralized citric acid: 10 wt%, $a_w \approx 0.975$) and subsequent washing. The thermograms are normalized with respect to their total integral. The average freezing onset temperatures are marked by black squares. For easier comparison the thermograms of the solutions are overlayed as dashed lines above the first thermogram. The displayed $T_{het}$ and $F_{het}$ values are the average of two measurements. The corresponding uncertainties are the standard deviation.





**Figure 7:** Freezing experiments with microcline (2 wt%) in aqueous polyol solutions as a function of water activity. Data points and curves are as explained in Fig. 3.







**Figure 8:** Freezing experiments with microcline (2 wt%) in aqueous amino acid solutions as a function of water activity. Data points and curves are as explained in Fig. 3.







**Figure 9:** DSC thermograms of microcline (2 wt%) in aqueous l-alanine solutions (1 wt%; $a_w \approx 0.994$) and b) aqueous l-lysine solutions (0.1 wt%; $a_w \approx 0.998$) measured directly after the preparation (fresh) and on the displayed days (day 1, 4, 7). Black squares mark the heterogeneous freezing onset temperatures. The thermograms are normalized with respect to their total integral. The displayed $T_{het}$ and $F_{het}$ values are the average of two measurements. The corresponding uncertainties are the standard deviation.





**Figure 10:** DSC thermograms of microcline (2 wt%) after 10 days of aging in the respective solutions (l-alanine: 10 wt%, $a_w \approx 0.977$; l-lysine: 1 wt%, $a_w \approx 0.994$) and subsequent washing. The thermograms are normalized with respect to their total integral. The average freezing onset temperatures are marked by black squares. For easier comparison the thermograms of the solutions are overlayed as dashed lines above the first thermogram. The displayed $T_{het}$ and $F_{het}$ values are the average of two measurements. The corresponding uncertainties are the standard deviation.





## Appendix

**Figure A1:** General reaction scheme of a zwitterionic equilibrium reaction of amino acids. In aqueous solution the $H^+$ of the carboxyl
770   function can be released and absorbed by the amino function of the same molecule leading to the formation of a so-called zwitterion. This
enables amino acids to strong polar interactions based on electrostatic forces.