# Peer review of "The impact of (bio-)organic substances on the ice nucleation activity of the K-feldspar microcline in aqueous solutions"

_Atmospheric Chemistry and Physics, 2021_

## Author Comment (AC2)

We thank Reviewer 2 for his/her positive feedback and the constructive comments. We reproduce reviewer's comments in blue and our responses in black.

Review of "the impact of (bio-) organic substances on the ice nucleation activity of the K-feldspar microcline in aqueous solutions" by Klumpp et al.

The authors used differential scanning calorimetry to investigate the ice-nucleating ability of K-feldspar in the presence of atmospherically relevant solutes, including carboxylic acids, amino acids, and polyols. Experiments were carried out as a function of solute concentration and exposure time. The effect of neutralization and washing of the K-feldspar after exposure was also studied to determine the importance of acid dissolution, anion binding, and adsorption. The results are convincing and the studies improve our understanding of the effect of solutes on ice nucleation by K-feldspar, an important atmospheric ice nucleus. The paper is well written, and as a result, I only have a few minor comments.

Minor comments:

- Section 2.3 the authors discuss experiments involving the neutralization of carboxylic acid solutions with ammonia. It was not clear at which stage the neutralization was carried out. Did they neutralize the acids before combining the acids with feldspar or after combining? Also, why did the authors neutralize with aqueous ammonia rather than use ammonium salts?

  We chose neutralization with ammonia to adjust the pH as close to a value of seven as possible. Yet, given the achieved precision of adjustment, using the ammonium salts would have been a valid alternative.

- Page 9, line 280: "Figure 5a shows no significant change of $T_{het}$ and $F_{het}$ over seven days when microcline is suspended in pure water". I think it would be worth mentioning at this point that although $T_{het}$ and $F_{het}$ were not significantly different, there was a clear change in the shape of the thermograms with the maximum of the heterogeneous freezing peaks shifting to lower temperatures.

  This is a good point and we checked the median freezing temperature as suggested by the reviewer. Please note that there is also variability in curve shape between replicate measurements as can be seen in the supplementary material. Taking the average of the measured thermograms, the median heterogeneous freezing temperature indeed decreases by 1.4 K due to aging, independent of the aging time. We mention this in the revised manuscript by adding (line 302–304):

  "*Figure 5a shows no significant change in $T_{het}$ and $F_{het}$ over seven days when microcline is suspended in pure water, yet, inspection of the thermograms reveals a shift of the median freezing temperature by about 1.4 K to lower temperature for the aged samples.*"

- Page 11, line 315. "the interaction of these molecules with the microcline surface must be very similar to the ones of water molecules, namely fast dynamic exchange of molecules at the microcline surface corresponding to the case shown in Fig. 1a." I think this statement would be more accurate if the authors replaced "microcline surface" with "ice-nucleating sites on the microcline surface" or something similar since their experiments are only sensitive to ice-nucleating sites, not the entire surface.

  This is a valid point. We change the sentence to:

*"…the interaction of these molecules with the ice-nucleating sites on the microcline surface must be very similar to the ones of water molecules, namely a fast dynamic exchange of molecules at the microcline surface corresponding to the case shown in Fig. 1a."*

---

## Author Comment (AC3)

We thank Reviewer 1 for his/her thoughtful comments. We reproduce the reviewer's comments in blue and our responses in black. Line numbers refer to the revised manuscript.

The paper reports on alteration of immersion freezing on microcline by several organic solutions. It is demonstrated that dissolution at the surface reduces the ice activity slowly and permanently, while complexation and adsorption reduce the ice activity rapidly and are reversible.

The paper is well written, experimental procedure and results are adequately rationalized. Suggestions for improvements are listed below. Once these are addressed the paper should be published in ACP.

Major comments:

1. By using the onset freezing temperature $T_{het}$ the effects on the few, most ice active microcline particles are highlighted. Onset freezing temperatures might not be the most relevant ice nucleation characteristic of a substances (as you mention on line 95f). Consider using the mean freezing temperature or the temperature of e.g., 10% frozen fraction as indicator for the bulk surface reaction to the treatments. Would using the mean freezing temperature for the analysis change the conclusions?

   Indeed, freezing onsets alone are not capable to fully capture the effect solutes exert on the IN activity of INPs. This is why we combine $F_{het}$ with $T_{het}$ as key parameters of freezing as observed in DSC thermograms. We have applied the same concept of combining $F_{het}$ with $T_{het}$ to characterize freezing curves in Kumar et al. (2018; 2019a, 2019b). In addition, we provide the full DSC thermograms in the supplementary material. We add an explanation of the concept of combining $F_{het}$ with $T_{het}$ to characterize heterogeneous freezing in the revised manuscript by extending the text starting on line 198:

   "*Our key parameters to analyze the thermograms are the onset temperatures of these peaks, $T_{het}$ and $T_{hom}$, together with the heterogeneously frozen fraction ($F_{het}$). We quantify $F_{het}$ by taking the integral over the portion of the thermogram between $T_{het}$ and $T_{hom}$ as shown in Fig. 2. The heat released during freezing is approximately proportional to the volume of water that freezes in the sample with a minor deviation from this proportionality arising from the temperature dependence of the freezing enthalpy (Speedy, 1987; Johari et al., 1994). Table 2 provides a summary of the uncertainty in each parameter. We consider a decrease/increase in $F_{het}$ as a loss/gain of IN active surface area. Assuming freezing on nucleation sites (Vali et al., 2015), this corresponds to the loss/generation of such sites. Conversely, a change in quality of the IN active surface area due to the deterioration/improvement of nucleation sites would lead to a decrease/increase in $T_{het}$. A concomitant decrease in $F_{het}$ is expected if some sites deteriorated to an extent that they cannot compete with homogeneous ice nucleation anymore. Since $T_{het}$ and $F_{het}$ are not able to capture the exact form of the heterogeneous freezing curve, we provide all DSC thermograms in the supplementary material. To derive the water activity from the melting point depression, the melting temperature ($T_{melt}$) is evaluated at the maximum of the melting peak.*"

2. There are two reasons why the reported frozen fractions $F_{het}$ change through the treatments. The number of freezing droplets changes and the range of freezing temperatures broadens to below the homogeneous freezing temperature. The first reason is due to the removal of activity, and the second due to a reduction of activity to various degree that leads to a broadening of the freezing temperature spectrum. Instead of discussing active sites (e.g., line 20ff), I propose to plot some, selected freezing temperature spectrum (frozen fraction as

function of temperature) to show and discuss the effect of treatments on the entire population of freezing droplets.

We agree that there are these two reasons for a change in $F_{het}$. We mention these two reasons now in the revised manuscript (see our reply above).

The DSC thermograms cannot be transformed unambiguously to frozen fraction, because the DSC does not register freezing events but the heat flow associated with the heat release during freezing. The relationship between freezing events and heat flow has been demonstrated in Marcolli et al. (2007) and depends on the heat release per time, which again depends on the cooling rate. Moreover, droplets within an emulsion are not monodisperse but exhibit a size distribution with consequences for the heat release per freezing event. This can be considered as a drawback of emulsion freezing experiments; conversely, emulsion freezing allows to compare different treatments effectively as it is relatively fast and gives a good overview of the overall freezing behavior. We will refer more often to the full thermograms shown in the supplementary information in the revised manuscript.

3. Please discuss the implications of the observed sensitivity of microcline to solutes on microclines assumed great importance for immersion freezing in the atmosphere. Is it not justified, or will cloud droplet activation prior to immersion freezing reverse most of the organic alterations?

This is a good point. We add the following discussion at the end of Sect. 4 "General discussion and implications" starting from line 440:

*"The high sensitivity to the presence of solutes gives rise to the question about the atmospheric relevance of microcline as an INP. Indeed, any investigated solute deteriorated the IN activity of microcline at water activities below 0.95. This is true for salts containing $Na^+$, $K^+$, and $NH_4^+$ as cations and $NO_3^-$, $SO_4^{2-}$, $Cl^-$ as anions, and even more for acidic solutions (Kumar et al., 2018; Whale et al., 2018; Yun et al., 2020; 2021). Moreover, also amino acids and neutralized dicarboxylic acids deteriorated the IN activity as shown in this study. This has consequences for the relevance of coated microcline particles as INPs at cirrus conditions. Even if the coating inhibits ice nucleation reversibly, IN activity would only be restored at RH >95 %, which is well above the RH where homogeneous ice nucleation in solution droplets sets in. Therefore, only uncoated microcline would be relevant as an INP at cirrus conditions. Conversely, microcline particles might act as INPs in immersion mode even when they are coated, if the coating inhibited ice nucleation reversibly, which is the case for salts and amino acids, but not for acidic coatings or coatings with dicarboxylic acids. While mineral dusts are uncoated at the source, they may obtain a coating during transport (Usher et al., 2003; Kolb et al., 2010; Ma et al., 2012; He et al., 2014; Fitzgerald et al., 2015; Tang et al., 2016). Even before long-range transport, soil dusts contain organic material, which may inhibit ice nucleation of microcline if present as coatings on the particles. To establish the role of coatings on the IN activity of microcline, soil dust and aged airborne dust would need to be collected, analyzed with respect to its mineralogical composition and investigated in controlled ice nucleation experiments."*

Specific comments:

1. The uncertainty in $F_{het}$ that is shown in Figs. 3,4,7,8 seems to underrepresent the observed variation between experiments. Looking at the data in the supplement, the frozen fraction in

repetitions of an experiments differ often by 10-20%. Consider showing the range of frozen fraction for all datapoints as vertical lines instead of averages.

We follow the reviewer's suggestion and show the freezing range of all data points in the revised manuscript.

2. Repeat the experiments marked with a black star in Figs. 3,4,7 to have at least one duplication.

As we explain in Sect. 2.1, we used two microcline stones from the same location, which are similar but not identical in composition and specific surface area. They are also similar but not altogether the same with respect to $F_{het}$ and $T_{het}$ as explained in Sect. 3.1.1. The first stone, which was the one of the study by Kumar et al. (2018), exhibited a quite sudden loss of IN activity within only a few weeks. We therefore had to change to a new stone. We decided not to mix experiments performed with different stones within one concentration series. This allows us to always reference to the case of dust from either the first or the second stone suspended in pure water. Because the experiments labeled with black stars have been performed with the first stone, it is not possible to repeat these measurements.

3. Section 2.2., line 174 ff. Explain why a readjustment of the procedure was necessary and how it affected the measurement.

The readjustment was required because the first rotor stator homogenizer broke at some point and we had to replace it including the dispersing aggregate. Although the replacement was the same model from the same company, the emulsions exhibited a slightly different size distribution and most importantly more large droplets leading to more spikes in the DSC thermograms. We therefore decided to readjust the preparation procedure of the emulsions. We add the following sentence on line 184 to motivate the readjustment of the procedure:

 *"This readjustment was required after the rotor stator had broken and was replaced by a new one of the same model (Polytron PT 1300D with a PT-DA 1307/2EC dispersing aggregate)."*

4. Line 184: clarify if every experiment was repeated or not. Other than stated here, the caption of Fig. 3 notes that some measurements were not repeated (marked with black stars).

We clarify in the revised manuscript on lines 191–192:

*"Every experiment was repeated at least once with a freshly prepared suspension unless stated otherwise (see figures and figure captions)".*

The experiments that were not repeated could not be repeated because the first stone lost most of its IN activity at some point as explained in response to specific comment #2.

5. Line 191: justify why onset temperatures are chosen to characterize freezing, but the peak maximum for melting.

We use the onset temperature $T_{het}$ together with $F_{het}$ to characterize the IN activity of the samples as explained in response to the major comment #1. We use the melting peak to determine the water activity of the solution by evaluating the melting point depression. For solutions, the melting peak is broadened due to eutectic melting and it is the maximum that represents the melting point depression. Therefore, it would be wrong to use the onset. We

mention now in the revised manuscript explicitly that we use the melting point to quantify the water activity (see response to major comment #1).

6. Line 243: The oxalic acid onset temperature data seem in disagreement to the results for neutralized oxalic acid shown in Fig. 4, which show a strong effect. The effects of the treatment could be shown clearer in a temperature spectrum of the frozen fraction.

The freezing onset of microcline in neutralized dilute oxalic acid solution is shifted to higher temperature because of the enhancing effect of ammonia. The same is the case for solutions with acetic and citric acid, as it is explained in the manuscript in Sect. 3.1.2. We do not see a disagreement. As explained before, conversion to frozen fraction is not possible.

7. Line 248: Instead of speculating about active sites, it could be stated that the treatment broadened the temperature range of droplet freezing and shifted it to lower temperatures. Both effects causing a larger fraction of droplets to freeze homogeneously.

We revise the text (lines 266–270) and refer to the form of the freezing curve:

*"However, the heterogeneous freezing curve of all carboxylic acids broadens and the median freezing temperature shifts to lower temperature (see Figs. S1–S3), which is manifested in a decrease in $F_{het}$ as shown in Fig. 3b and points to a loss of IN activity in acid solution (high $H^+$ concentration). The same was observed by Kumar et al. …."*

8. Line 282f: Looking at the shift of the peak heat flow in Fig. 5a), a decrease in frozen fraction seems often accompanied by a decrease in the median freezing temperature similar to the 2K reported in Peckhaus et al. 2016. Please check if the shift in median temperature does occur and is only not present in the onset temperature.

This is a good point and we checked the median freezing temperature as suggested by the reviewer. Please note that there is also variability in curve shape between replicate measurements as can be seen in the supplementary material. Taking the average of the measured thermograms, the median heterogeneous freezing temperature indeed decreases by 1.4 K due to aging, independent of the aging time. We mention this in the revised manuscript (lines 302–304):

*"Figure 5a shows no significant change in $T_{het}$ and $F_{het}$ over seven days when microcline is suspended in pure water, yet, inspection of the thermograms reveals a shift of the median freezing temperature by about 1.4 K to lower temperature for the aged samples."*

9. Line 394ff: Can you propose a method to test your suggestion that reduced ice formation after heat or $H_2O_2$ treatment is due to removing ammonia? Heat sensitivity of samples from remote locations could be counterevidence to your suggestion.

A simple method to test this suggestion would be to perform TGA-MS analysis with the samples before heat or $H_2O_2$ treatment to determine the effective content of organics or ammonia. Whether loss of ammonia could be detected might also depend on the measurement protocol. If the samples are dried strongly before starting the heating ramp, ammonia might be lost. An additional test would be to perform again a TGA-MS with the $H_2O_2$ or heat-treated samples to quantify to what extent the organics have been removed.

10. Fig. 1d): Explain how dissolution reduces the ice formation activity. Does it come to a reprecipitation and coating of the microcline particles with a secondary mineral after dissolution?

We add an explanation of the effect dissolution has on the IN activity in the revised manuscript on lines 131–135:

[revised manuscript text omitted]